# Correction of the Scientific Production: Publisher Performance Evaluation Using a Dataset of 4844 PubMed Retractions

**Catalin Toma** [1,*] **, Liliana Padureanu** [2] **and Bogdan Toma** [3]

1 Independent Researcher, 700054 Iasi, Romania
2 Department of Immunology and Alergology, Emergency Clinical Hospital 'St. Spiridon', 700098 Iasi, Romania; lilianapadureanu@gmail.com
3 Faculty of Medicine, "Grigore T. Popa" University of Medicine and Pharmacy of Iași, 700098 Iasi, Romania; bogdan.toma808@gmail.com
* Correspondence: catalin.toma@gmail.com

**Abstract: Background.** Retraction of problematic scientific articles after publication is one of the mechanisms for correcting the literature available to publishers. The market volume and the business model justify publishers' ethical involvement in the post-publication quality control (PPQC) of human-health-related articles. The limited information about this subject led us to analyze Pub-Med-retracted articles and the main retraction reasons grouped by publisher. We propose a score to appraise publisher's PPQC results. The dataset used for this article consists of 4844 Pub-Med-retracted papers published between 1.01.2009 and 31.12.2020. **Methods.** An SDTP score was constructed from the dataset. The calculation formula includes several parameters: speed (article exposure time (ET)), detection rate (percentage of articles whose retraction is initiated by the edi-tor/publisher/institution without the authors' participation), transparency (percentage of retracted articles available online and the clarity of the retraction notes), and precision (mention of authors' responsibility and percentage of retractions for reasons other than editorial errors). **Results.** The 4844 retracted articles were published in 1767 journals by 366 publishers, the average number of retracted articles/journal being 2.74. Forty-five publishers have more than 10 retracted articles, holding 88% of all papers and 79% of journals. Combining our data with data from another study shows that less than 7% of PubMed dataset journals retracted at least one article. Only 10.5% of the retraction notes included the individual responsibility of the authors. Nine of the top 11 publishers had the largest number of retracted articles in 2020. Retraction-reason analysis shows considerable differences between publishers concerning the articles' ET: median values between 9 and 43 months (mistakes), 9 and 73 months (images), and 10 and 42 months (plagiarism and overlap). The SDTP score shows, from 2018 to 2020, an improvement in PPQC of four publishers in the top 11 and a decrease in the gap between 1st and 11th place. The group of the other 355 publishers also has a positive evolution of the SDTP score. **Conclusions.** Publishers have to get involved actively and measurably in the post-publication evaluation of scientific products. The introduction of reporting standards for retraction notes and replicable indicators for quantifying publishing QC can help increase the overall quality of scientific literature.

**Keywords:** PubMed retractions; scientific publishing; quality control; retraction notes; retraction reporting; publishers

## 1. Introduction

*"One of the greatest criticisms in the blogosphere is not so much that the current rules and guidelines are weak or poor, but that enforcement and irregular application of those rules, particularly by COPE member journals and publishers, confuses the readership, disenfranchises authors who remain confused (despite having a stricter and more regulated system) and provides an imbalanced publishing structure that has weak, or limited, accountability or transparency."* [1]

The publication of scientific literature represents, globally, a market of considerable size, which reached a record value of $28 billion in 2019 (from $9.4 billion in 2011 [2]) and fell to $26.5 billion in 2020, with forecasts suggesting a recovery of losses by 2023. Revenues from the publication of articles in 2019 were $10.81 billion and those from the publication of books $3.19 billion, with derivative products representing the difference. The segment of medical publications ($12.8 billion in 2020) is constantly growing. Estimations show that in 2024 the medical literature will exceed the volume of the technical and scientific literature [3]. The continued growth was accompanied by a consolidation process that made the top five publishers in 2013 represent over 50% of all published articles. These changes occur in an atypical market where publishers have high profit margins [4], do not pay for purchased goods (authors are not paid), do not pay for quality control (peer review), and have a monopoly on the content of published articles (Ingelfinger law) [5]. In the case of medical literature, because of the direct impact that scientific errors or fraud may have on patients' health, publishers have an ethical obligation to invest resources in post-publication quality control and disseminate identified issues as quickly and widely as possible. One of the methods is the retraction of scientifically, ethically, or legally invalid articles (questionable research practices (QRP), questionable publication practices (QPP)). The interest in correcting the literature seems to be confirmed by recent developments: the number of journals with at least one retracted article increased from 44 in 1997 to 488 in 2016 [6]. The continued growth may be due to an improved capacity to detect and remove problematic articles [7], which, despite a somewhat reluctant if not a resisting editorial environment [8,9] and the lack of significant progress in reporting [1,10] continues to expand in the publishing environment. For example, in the case of the PubMed retractions, 2020 was a record year in terms of retraction notes, targeting 878 articles published in more than 12 years [11].

The retractions in the biomedical journals indexed in PubMed represent a topic that was intensely researched in the last two decades, numerous articles having made valuable contributions in this field: the number of authors [12,13], author countries [14–16], publication types [12,13,17], exposure time/time to retraction, retraction reasons, citations received by retracted articles, financing of retracted research, impact factor of journals publishing retracted research, who is retracting, study types, and retraction responsibility [12–25]. However, there is little information on the PubMed article retractions at the publisher level and, therefore, an incomplete picture of the challenges/difficulties they face in the post-publication quality control of products delivered to consumers of scientific information. Several authors point out issues that arise when retracting a scientific article:

– The process of retracting an article is a complex one, depending on several factors: who initiates the retraction, the context, the communication between the parties, and the editorial experience [26].
– The clarity of the retraction notes leaves much to be desired and presents a significant variability between journals and/or publishers in relation to the COPE guidelines [1,27–29] although a uniform approach has long been required [15].
– The individual contribution of the authors is rarely mentioned in the retraction notes, contrary to the COPE recommendations [29].
– The online presence of retracted articles, required in the COPE guidelines, ensures more transparency and avoids the occurrence of "silent or stealth retractions" [30].
– The role of publishers and editors avoiding what is called editorial misconduct is an important one [31]. Editorial errors (duplicate publication, accidental publication of wrong version/rejected article, wrong journal publication) were identified in different proportions: 7.3% (328 cases) in a 2012 study that analyzed 4449 articles retracted between 1928–2011 [32], 1.5% (5 cases) in another study [29], 5% in the Bar-Ilan study [33], and 3.7% in our study on PubMed retractions between 2009 and 2020 [11].

- The level of involvement of publishers and editors in article retraction is variable [28,32] although most of them can initiate the retraction of an article without the authors' consent [34].
- The different efficiencies of the QRP and QPP detection mechanisms at the editorial level may explain the differences between publishers [35]; the possible application of post-publication peer review [36,37] can contribute both to the increase of the detection capacity and the reduction of the differences between journals/publishers.

Editorial policies are/can be implemented at the publisher level and could positively/negatively affect the performance of all journals in its portfolio. We considered it would be constructive to present an overview of retracted articles and a structure of retraction reasons of major publishers using a dataset obtained from the analysis of 4844 biomedical PubMed retractions from the period 2009–2020.

We also deemed it worthwhile to initiate a debate on the performance of publishers in correcting the scientific literature. For this reason, we propose a score based on four indicators: speed of article retraction, post-publication ability of the publisher/editors to detect QRP/QPP articles, transparency (measured by the online maintenance of retracted articles and the clarity of retraction notes), and the precision of the correction process (identification of those responsible and the degree of avoidance of editorial errors).

Our study has three main objectives:

1. Current numbers and progression of 2009–2020 human-health-related PubMed retracted articles and their retraction notes, grouped by publisher.
2. Retraction reasons for publishers on the first 10 positions.
3. Appraisal of publisher quality control and post-publication peer review effectiveness using a score based on several components.

What do we consider new in this article?

Presentation of the main retraction reasons and exposure time (ET) for leading publishers:

- Dynamics of retraction notes at publisher level;
- Evaluation of publisher performance for the three main retraction reasons;
- Proposal of a score for measuring publisher post-publication QC performance (SDTP score: speed-detection-transparency-precision score)

## 2. Materials and Methods

The methodology used to collect the data is presented in detail in another article [11]. Here we describe the main elements: analyzed period: 2009–2020 (last update for retracted articles made on 31 January 2021); data source for retracted articles: PubMed csv files; source of data for publishers holding journals in which the retracted articles and notes were published: SCOPUS; and the period during which the data were analyzed: July 2020–May 2021.

Information extraction and partial processing: teamwork web application developed by the first author in which PubMed csv files were imported and analyzed.

Final processing: export from the web application to the SPSS, coding, and analysis.

For the publishing and editorial performance indicators, we have built a score (SDTP score, Table 1) consisting of four components and six values represented equally and calculated from the dataset collected: speed, detection rate, transparency, and precision.

**Table 1.** SDTP score components.

| | | | Component and Rationale for Inclusion |
|---|---|---|---|
| **1** | **Speed** | | Speed of retraction, measured in months [38] |
| **2** | **Detection rate** | | Percentage of total retracted articles for which the retraction was initiated by/involved the editor, the editorial board, the publisher, or institutions, without authors [38] |
| **3** | **Transparency** | a | The percentage of papers available online ("Retracted articles should not be removed from electronic archives or printed copies of the journal." [39]). |
| | | b | The retraction note contains the reasons [38] |
| **4** | **Precision** | a | Individual responsibility of authors was clearly stated [38]. Articles with more than one author and no editorial errors were analyzed. |
| | | b | Percentage of retractions for reasons other than editorial errors (not attributable to editorial errors). |

In the case of publisher/editor involvement (2-Detection rate), we used in the calculation all articles that in the retraction note mentioned the involvement of the publisher, editor in chief, editorial board, institution, and Office of Research Investigation, without authors.

To measure the identification of the authors' responsibility (5-Precision-Individual responsibility), we used as the calculation base the number of articles with more than one author and retractions for other reasons than editorial errors.

*Calculation of SDTP Score*

The values for each publisher are compared with the values of the entire set of 4844 retracted articles (2009–2020), 3931 articles (2009–2019), and 3361 articles (2009–2018). There are two situations:

A. Values above average are considered poor performance. Examples: exposure time (speed); calculation formula for speed: 100 − (publisher value/baseline value) * 100]
B. A percentage value above the average is considered a good performance. Example: detection rate, percentage of online papers, percentage of clear retraction notes, percentage of retractions in which individual author responsibilities are mentioned, percentage of retractions that are not due to editorial errors. Formula: [(publisher value/baseline value) * 100] − 100.

The values obtained are summed and form the SDTP score of the publisher.

## 3. Results

### 3.1. Retractions by Publishers

The 4844 retracted articles were published in 1767 journals. The average number of retracted articles is 2.74/journal. Several studies have reported publisher rankings, with the top positions being consistently occupied by publishers with a large number of publications [7,28,40]. This is also reflected in the results obtained by us. Forty-five publishers with more than 10 retractions account for 88% (n = 4261) of retractions and 79% (n = 1401) of journals (Table 2). The remaining 583 papers are published in 366 journals associated with 321 publishers.

**Table 2.** Publishers with more than 10 retracted articles.

| | Publisher | Articles | Journals | /Journal |
|---|---|---|---|---|
| 1 | ELSEVIER | 846 | 309 | 2.74 |
| 2 | SPRINGER NATURE | 749 | 305 | 2.46 |
| 3 | WOLTERS KLUWER | 346 | 142 | 2.45 |
| 4 | WILEY-BLACKWELL | 325 | 171 | 1.9 |
| 5 | PLOS | 275 | 7 | 39.28 |
| 6 | SAGE | 248 | 47 | 5.28 |
| 7 | TAYLOR AND FRANCIS | 223 | 95 | 2.35 |
| 8 | HINDAWI | 140 | 52 | 2.69 |
| 9 | DOVE MEDICAL PRESS | 111 | 23 | 4.83 |
| 10 | E-CENTURY PUBLISHING | 71 | 7 | 10.14 |
| 11 | SPANDIDOS PUBLICATIONS | 71 | 7 | 10.14 |
| 12 | OXFORD UNIVERSITY PRESS | 65 | 39 | 1.67 |
| 13 | MARY ANN LIEBERT | 63 | 26 | 2.42 |
| 14 | VERDUCI EDITORE | 56 | 1 | 56 |
| 15 | AMERICAN CHEMICAL SOCIETY | 53 | 17 | 3.12 |
| 16 | AMERICAN ASSOCIATION FOR CANCER RESEARCH | 49 | 6 | 8.17 |
| 17 | AMERICAN SOCIETY FOR MICROBIOLOGY | 44 | 10 | 4.4 |
| 18 | FRONTIERS MEDIA | 44 | 18 | 2.44 |
| 19 | MULTIDISCIPLINARY DIGITAL PUBLISHING INSTITUTE | 42 | 17 | 2.47 |
| 20 | NATIONAL ACADEMY OF SCIENCES | 36 | 1 | 36 |
| 21 | PORTLAND PRESS | 32 | 3 | 10.67 |
| 22 | BMJ PUBLISHING GROUP | 29 | 17 | 1.7 |
| 23 | AMERICAN PHYSIOLOGICAL SOCIETY | 26 | 8 | 3.25 |
| 24 | FUNDACAO DE PESQUISAS CIENTIFICAS DE RIBEIRAO PRETO | 20 | 1 | 20 |
| 25 | AMERICAN MEDICAL ASSOCIATION | 19 | 12 | 1.58 |
| 26 | SOCIETY FOR NEUROSCIENCE | 18 | 1 | 18 |
| 27 | AMERICAN ASSOCIATION FOR THE ADVANCEMENT OF SCIENCE | 17 | 3 | 5.66 |
| 28 | AMERICAN SOCIETY OF HEMATOLOGY | 17 | 1 | 17 |
| 29 | AMERICAN SOCIETY FOR PHARMACOLOGY AND EXPERIMENTAL THERAPEUTICS | 16 | 3 | 5.33 |
| 30 | BENTHAM | 16 | 8 | 2 |
| 31 | CELL PHYSIOL BIOCHEM PRESS | 16 | 1 | 16 |
| 32 | IMPACT JOURNALS | 16 | 2 | 8 |
| 33 | MEDICAL SCIENCE INTERNATIONAL PUBLISHING | 16 | 1 | 16 |
| 34 | AMERICAN DIABETES ASSOCIATION | 15 | 2 | 7.5 |
| 35 | AMERICAN SOCIETY OF CLINICAL INVESTIGATION | 15 | 1 | 15 |
| 36 | CUREUS, INC. | 14 | 1 | 14 |
| 37 | THE COMPANY OF BIOLOGISTS LTD. | 13 | 2 | 6.5 |
| 38 | KOWSAR PUBLISHING COMPANY | 12 | 3 | 4 |
| 39 | KARGER | 12 | 8 | 1.5 |
| 40 | AME PUBLISHING COMPANY | 11 | 4 | 2.75 |
| 41 | AMERICAN THORACIC SOCIETY | 11 | 3 | 3.66 |
| 42 | ASSOCIACAO BRASILEIRA DE DIVULGACAO CIENTIFICA | 11 | 1 | 11 |
| 43 | FUTURE MEDICINE LTD | 11 | 9 | 1.22 |
| 44 | INTERNATIONAL INSTITUTE OF ANTICANCER RESEARCH | 11 | 1 | 11 |
| 45 | IOS Press | 10 | 5 | 2 |

The top 11 publishers have 3405 retracted articles (70.3%) in 1165 journals (65.9%). In the following, we will only analyze their evolution and performance. The rest of the publishers will be analyzed within a single group.

### 3.2. Retraction Notes/Publisher (2009–2020)

The year 2020 is the most consistent year for retracted articles for almost all publishers in the top 11, except PLOS, which peaked in 2019, SAGE in 2017, and E-Century Publishing in 2015 (Table 3). The year 2015 seems to be, for most publishers, the beginning of a greater interest in correcting the medical literature.

**Table 3.** Retracted articles and retraction notes by year for top 11 publishers.

| Publisher | N | 2009 | 2010 | 2011 | 2012 | 2013 | 2014 | 2015 | 2016 | 2017 | 2018 | 2019 | 2020 | 2021 |
|---|---|---|---|---|---|---|---|---|---|---|---|---|---|---|
| **ELSEVIER** | 846 | 68 | 87 | 82 | 111 | 82 | 77 | 70 | 63 | 70 | 68 | 47 | 21 | |
| **Retraction Notes** | | 6 | 15 | 38 | 42 | 70 | 53 | 78 | 100 | 84 | 116 | 112 | 129 | 3 |
| **SPRINGER NATURE** | 751 | 48 | 64 | 46 | 68 | 74 | 116 | 87 | 69 | 49 | 47 | 51 | 32 | |
| **Retraction Notes** | RN | 8 | 18 | 37 | 45 | 52 | 43 | 123 | 94 | 61 | 58 | 73 | 127 | 12 |
| **WOLTERS KLUWER** | 346 | 33 | 34 | 33 | 34 | 32 | 40 | 27 | 33 | 27 | 19 | 20 | 14 | |
| **Retraction Notes** | RN | 4 | 12 | 22 | 26 | 21 | 41 | 32 | 40 | 28 | 30 | 44 | 46 | 0 |
| **WILEY-BLACKWELL** | 323 | 44 | 40 | 26 | 39 | 30 | 24 | 30 | 17 | 15 | 18 | 19 | 21 | |
| **Retraction Notes** | RN | 12 | 15 | 21 | 24 | 23 | 26 | 22 | 37 | 24 | 41 | 18 | 56 | 4 |
| **PLOS** | 275 | 8 | 11 | 28 | 44 | 64 | 56 | 25 | 11 | 11 | 6 | 8 | 3 | |
| **Retraction Notes** | RN | 1 | 2 | 0 | 5 | 9 | 7 | 7 | 15 | 18 | 43 | 92 | 73 | 3 |
| **SAGE** | 248 | 5 | 6 | 7 | 12 | 25 | 60 | 67 | 28 | 7 | 11 | 11 | 9 | |
| **Retraction Notes** | RN | 0 | 0 | 8 | 7 | 2 | 12 | 40 | 21 | 119 | 9 | 11 | 19 | 0 |
| **TAYLOR AND FRANCIS** | 223 | 19 | 13 | 32 | 21 | 18 | 14 | 10 | 15 | 8 | 15 | 40 | 18 | |
| **Retraction Notes** | RN | 6 | 3 | 9 | 36 | 11 | 18 | 12 | 21 | 17 | 5 | 21 | 59 | 5 |
| **HINDAWI** | 140 | 1 | 2 | 9 | 15 | 22 | 37 | 25 | 15 | 5 | 6 | 3 | 0 | |
| **Retraction Notes** | RN | 0 | 0 | 0 | 0 | 6 | 17 | 14 | 21 | 17 | 14 | 20 | 30 | 1 |
| **DOVE MEDICAL PRESS** | 111 | 2 | 4 | 10 | 11 | 3 | 7 | 9 | 8 | 8 | 12 | 17 | 20 | |
| **Retraction Notes** | RN | 0 | 0 | 0 | 5 | 2 | 4 | 3 | 11 | 11 | 6 | 14 | 54 | 1 |
| **E-CENTURY PUBLISHING** | 71 | **0** | **0** | 0 | **0** | 2 | 8 | 34 | 6 | **6** | 3 | 8 | 4 | |
| **Retraction Notes** | RN | 0 | 0 | 0 | 0 | 1 | 0 | 9 | 24 | 6 | 5 | 9 | 17 | 0 |
| **SPANDIDOS PUBLICATIONS** | 71 | **0** | **0** | 1 | **5** | 5 | 6 | 7 | 18 | **15** | 11 | 0 | 3 | |
| **Retraction Notes** | RN | 0 | 0 | 0 | 2 | 1 | 4 | 5 | 11 | 7 | 13 | 10 | 18 | 0 |
| **RN for top publishers** | | 37 | 65 | 135 | 192 | 198 | 225 | 345 | 395 | 392 | 340 | 424 | 628 | |
| **Total number of RN** | | 54 | 97 | 195 | 282 | 300 | 337 | 481 | 570 | 544 | 501 | 570 | 878 | |

The top 11 publishers hold 70.3% (3405/4844) of retracted articles and 66% (1165/1767) of journals. The annual distribution of retraction notes varies between 66% of the total (2013) and 74% (2019).

### 3.3. Publishers and Retraction Reasons

The retraction reasons of the top 11 publishers are presented in Table 4. Multiple reasons in one retraction note were added to the respective categories, thus explaining the publisher percentage sums higher than 100%.

**Table 4.** Retraction reasons for top 11 publishers (3405 retracted papers, 70.3%).

| Publisher | | MISTAKES | IMAGES | PLAGIARISM | OVERLAP | FRAUD | ETHICS | AUTHORSHIP | UNCLEAR | EDITOR | PROPERTY | OTHER |
|---|---|---|---|---|---|---|---|---|---|---|---|---|
| ELSEVIER | 846 | 32.98% | 33.2% | 9.8% | 10.87% | 2.6% | 6.38% | 4.73% | 5.79% | 4.25% | 1.3% | 1.3% |
| | N | 279 | 281 | 83 | 92 | 22 | 54 | 40 | 49 | 36 | 11 | 11 |
| SPRINGER NATURE | 749 | 29.64% | 17.62% | 18.42% | 12.02% | 17.09% | 8.94% | 5.74% | 0.5% | 2.4% | 4.67% | 1.07% |
| | N | 222 | 132 | 138 | 90 | 128 | 67 | 43 | 4 | 18 | 35 | 8 |
| WOLTERS KLUWER | 346 | 28.61% | 6.94% | 24.85% | 15.6% | 1.73% | 8.67% | 6.36% | 6.65% | 6.94% | 1.73% | 1.73% |
| | N | 99 | 24 | 86 | 54 | 6 | 30 | 22 | 23 | 24 | 6 | 6 |
| WILEY-BLACKWELL | 325 | 36% | 18.8% | 9.2% | 14.5% | 1.5% | 9.5% | 7.1% | 2.2% | 6.5% | 2.2% | 1.8% |
| | N | 117 | 61 | 30 | 47 | 5 | 31 | 23 | 7 | 21 | 7 | 6 |
| PLOS | 275 | 29.5% | 63.3% | 4% | 4% | 1.1% | 16.4% | 5.1% | – | 0.7% | 1.1% | 0.4% |
| | N | 81 | 174 | 11 | 11 | 3 | 45 | 14 | – | 2 | 3 | 1 |
| SAGE | 248 | 16.9% | 0.8% | 9.3% | 8.1% | 62.9% | 4.8% | 7.3% | 1.2% | 2.4% | 1.2% | – |
| | N | 42 | 2 | 23 | 20 | 156 | 12 | 18 | 3 | 6 | 3 | – |
| TAYLOR AND FRANCIS | 223 | 26.9% | 11.7% | 13% | 17% | 13% | 6.7% | 10.3% | 1.8% | 6.7% | 2.7% | 1.3% |
| | N | 60 | 26 | 29 | 38 | 29 | 15 | 23 | 4 | 15 | 6 | 3 |
| HINDAWI | 140 | 22.1% | 27.1% | 33.6% | 16.4% | 0.7% | 9.3% | 10% | 0.7% | – | 2.1% | – |
| | N | 31 | 38 | 47 | 23 | 1 | 13 | 14 | 1 | – | 3 | – |
| DOVE MEDICAL PRESS | 111 | 27.9% | 39.6% | 14.4% | 16.2% | 2.7% | 14.4% | 4.5% | – | – | 1.8% | – |
| | N | 31 | 44 | 16 | 18 | 3 | 16 | 5 | – | – | 2 | – |
| E-CENTURY PUBLISHING | 71 | 26.8% | 2.8% | 36.6% | 1.4% | 8.5% | 2.8% | 4.2% | – | 25.4% | 1.4% | – |
| | N | 19 | 2 | 26 | 1 | 6 | 2 | 3 | – | 18 | 1 | – |
| SPANDIDOS PUBLICATIONS | 71 | 33.8% | 29.6% | 28.2% | 5.6% | 1.4% | 7% | 11.3% | – | – | 1.4% | 1.4% |
| | N | 24 | 21 | 20 | 4 | 1 | 5 | 8 | – | – | 1 | 1 |

*3.4. Mistakes/Inconsistent Data*

Of the 1553 cases, 1005 (64.7%) belong to the top 11 publishers, presented in Table 5. The rest of the publishers account for 548/1553 cases. In 229/1553 cases (14.7%, 95% CI 12.9–16.5), the reason for the retraction was data fabrication. For the top 11 publishers, there are 127/1005 (12.6%, CI95% 12.6–14.7) cases of data fabrication (48 Elsevier, 22 for Springer Nature and Wolters Kluwer, 12 for Wiley-Blackwell, 7 for SAGE, 5 for E-Century Publishing, 4 for Taylor & Francis and PLOS, and 3 for Hindawi). The other publishers have 102/548 articles retracted for data fabrication (18.6%, CI95% 15.3–21.8).

**Table 5.** Mistakes/Inconsistent data per publisher.

| | Articles | ET (95% CI) | Median | Range | IQR |
|---|---|---|---|---|---|
| Elsevier | 279 | 29.03 (25.87–32.19) | 20 | 133 | 33 |
| Springer Nature | 222 | 27.76 (24.04–31.48) | 17 | 140 | 29 |
| Wolters Kluwer | 99 | 19.69 (15.26–24.12) | 11 | 99 | 16 |
| Wiley-Blackwell | 117 | 26.19 (22.45–29.93) | 23 | 98 | 26 |
| PLOS | 81 | 41.58 (34.92–48.24) | 43 | 117 | 57 |
| SAGE | 42 | 21.36 (16.05–26.67) | 17 | 61 | 27 |
| Taylor & Francis | 60 | 15.43 (11.34–19.53) | 12 | 72 | 16 |
| Hindawi | 31 | 29.74 (20.24–39.25) | 16 | 81 | 41 |
| Dove Medical Press | 31 | 31.90 (16.62–47.19) | 9 | 139 | 46 |
| E-Century Publishing | 19 | 22.58 (15.48–29.68) | 26 | 48 | 26 |
| Spandidos Publications | 24 | 17.17 (9.68–24.65) | 9 | 70 | 20 |

The lowest ET average belongs to Taylor & Francis (15.4 months) and the highest to PLOS (41.5 months). We note in the meantime that in most of the cases skewed distributions and median values ranging from an encouraging 9 months (Dove Medical Press and Spandidos Publications), 11 months (Wolters Kluwer), and 12 months (Taylor & Francis) to a rather unexpected 43 months for PLOS.

### 3.5. Images

Several publishers have a high number of retractions that were due to image problems: PLOS (174 of 275 papers, 63.3%), Elsevier (281 of 846 papers, 33.2%), and Springer Nature (132 of 749 papers, 17.62%), possibly signaling the implementation of procedures and technologies to detect problematic articles.

In the case of PLOS (details presented in Table 6), out of the 174 articles retracted for image problems, 150 (86.2%) were published between 2011–2015 and 90.9% (158) of the retraction notes were published between 2017–2020. This suggests that 2017 could be the year when the systematic and retroactive verification of the images in the articles published in 2009–2020 began. Only 10 articles published by PLOS between 2016–2020 (no articles in 2019 and 2020) were retracted because of images, suggesting the effectiveness of the measures implemented by this publisher and that, probably, the articles with questionable images were stopped before publication.

**Table 6.** Evolution of the PLOS image retractions.

| | | Retraction Note Year | | | | | | | | | | Total |
| | | 2010 | 2013 | 2014 | 2015 | 2016 | 2017 | 2018 | 2019 | 2020 | 2021 | |
|---|---|---|---|---|---|---|---|---|---|---|---|---|
| | 2009 | 0 | 0 | 0 | 0 | 0 | 0 | 0 | 2 | 2 | 0 | 4 |
| | 2010 | 1 | 0 | 0 | 0 | 1 | 0 | 2 | 2 | 4 | 0 | 10 |
| | 2011 | 0 | 1 | 0 | 0 | 2 | 1 | 7 | 5 | 5 | 0 | 21 |
| | 2012 | 0 | 2 | 0 | 0 | 0 | 2 | 2 | 11 | 10 | 1 | 28 |
| | 2013 | 0 | 0 | 1 | 2 | 0 | 3 | 8 | 11 | 21 | 2 | 48 |
| Publication Year | 2014 | 0 | 0 | 0 | 0 | 2 | 4 | 7 | 17 | 12 | 0 | 42 |
| | 2015 | 0 | 0 | 0 | 1 | 0 | 2 | 1 | 3 | 4 | 0 | 11 |
| | 2016 | 0 | 0 | 0 | 0 | 0 | 0 | 0 | 3 | 1 | 0 | 4 |
| | 2017 | 0 | 0 | 0 | 0 | 0 | 0 | 1 | 1 | 1 | 0 | 3 |
| | 2018 | 0 | 0 | 0 | 0 | 0 | 0 | 1 | 1 | 1 | 0 | 3 |
| Total | | 1 | 3 | 1 | 3 | 5 | 12 | 29 | 56 | 61 | 3 | 174 |

In the case of Elsevier, out of the total of 281 articles retracted for image issues, 246 (87.5%) were retracted between 2015–2020 (2016 is the first year with a significant number of retraction notes, almost half of those published in that year) and the period 2016–2020 was characterized by a slight decrease in problematic articles (74, 26.3%). Elsevier did not have any articles retracted in 2020 because of image problems. These data suggest an increased effectiveness in dealing with image issues. The year 2016 seems to mark the beginning of implementing the procedures and technologies for image analysis at this publisher (Table 7).

**Table 7.** Elsevier image retractions.

| | | Retraction Note Year | | | | | | | | | | | | Total |
| | | 2010 | 2011 | 2012 | 2013 | 2014 | 2015 | 2016 | 2017 | 2018 | 2019 | 2020 | 2021 | |
|---|---|---|---|---|---|---|---|---|---|---|---|---|---|---|
| | 2009 | 1 | 1 | 2 | 1 | 1 | 1 | 4 | 3 | 3 | 2 | 2 | 0 | 21 |
| | 2010 | 1 | 1 | 2 | 2 | 3 | 3 | 16 | 3 | 1 | 1 | 2 | 0 | 35 |
| | 2011 | 0 | 0 | 0 | 3 | 3 | 4 | 7 | 4 | 4 | 6 | 4 | 0 | 35 |
| | 2012 | 0 | 0 | 2 | 3 | 1 | 1 | 10 | 7 | 3 | 3 | 5 | 0 | 35 |
| | 2013 | 0 | 0 | 0 | 2 | 3 | 5 | 2 | 6 | 2 | 2 | 7 | 0 | 29 |
| Publication Year | 2014 | 0 | 0 | 0 | 0 | 1 | 3 | 2 | 5 | 5 | 4 | 4 | 0 | 24 |
| | 2015 | 0 | 0 | 0 | 0 | 0 | 1 | 4 | 4 | 6 | 8 | 5 | 0 | 28 |
| | 2016 | 0 | 0 | 0 | 0 | 0 | 0 | 2 | 5 | 4 | 10 | 5 | 0 | 26 |
| | 2017 | 0 | 0 | 0 | 0 | 0 | 0 | 0 | 2 | 6 | 7 | 3 | 1 | 19 |
| | 2018 | 0 | 0 | 0 | 0 | 0 | 0 | 0 | 0 | 2 | 6 | 8 | 1 | 17 |
| | 2019 | 0 | 0 | 0 | 0 | 0 | 0 | 0 | 0 | 0 | 1 | 11 | 0 | 12 |
| Total | | 2 | 2 | 6 | 11 | 12 | 18 | 47 | 39 | 36 | 50 | 56 | 2 | 281 |

In the case of Springer Nature, the focus on images manifested itself a little later (2018–2019), with each year from 2009–2020 containing articles retracted because of the images (Table 8).

**Table 8.** Springer Nature image retractions.

| | | Retraction Note Year | | | | | | | | | | | | Total |
|---|---|---|---|---|---|---|---|---|---|---|---|---|---|---|
| | | 2009 | 2011 | 2012 | 2013 | 2014 | 2015 | 2016 | 2017 | 2018 | 2019 | 2020 | 2021 | |
| | 2009 | 1 | 3 | 1 | 1 | 0 | 0 | 1 | 0 | 0 | 0 | 0 | 1 | 8 |
| | 2010 | 0 | 2 | 2 | 1 | 1 | 0 | 1 | 1 | 0 | 0 | 1 | 0 | 9 |
| | 2011 | 0 | 2 | 1 | 2 | 0 | 0 | 1 | 1 | 2 | 0 | 0 | 0 | 9 |
| | 2012 | 0 | 0 | 1 | 2 | 2 | 0 | 1 | 1 | 0 | 2 | 1 | 0 | 10 |
| | 2013 | 0 | 0 | 0 | 1 | 0 | 2 | 1 | 0 | 0 | 1 | 2 | 1 | 8 |
| Publication Year | 2014 | 0 | 0 | 0 | 0 | 2 | 1 | 1 | 1 | 4 | 1 | 4 | 0 | 14 |
| | 2015 | 0 | 0 | 0 | 0 | 0 | 0 | 2 | 3 | 3 | 2 | 2 | 0 | 12 |
| | 2016 | 0 | 0 | 0 | 0 | 0 | 0 | 1 | 3 | 2 | 9 | 1 | 1 | 17 |
| | 2017 | 0 | 0 | 0 | 0 | 0 | 0 | 0 | 0 | 2 | 5 | 10 | 1 | 18 |
| | 2018 | 0 | 0 | 0 | 0 | 0 | 0 | 0 | 0 | 0 | 5 | 2 | 0 | 7 |
| | 2019 | 0 | 0 | 0 | 0 | 0 | 0 | 0 | 0 | 0 | 2 | 13 | 1 | 16 |
| | 2020 | 0 | 0 | 0 | 0 | 0 | 0 | 0 | 0 | 0 | 0 | 4 | 0 | 4 |
| Total | | 1 | 7 | 5 | 7 | 5 | 3 | 9 | 10 | 13 | 27 | 40 | 5 | 132 |

Exposure time (ET) for articles with image problems (Table 9) varies between median values of 9/11 months (Dove Medical Press/Spandidos Publications) and 73 months (PLOS). For most other publishers, the median values are between 43 and 63.5 months (average ET value is over 50 months), an exception being Springer Nature, with a median value of 28 months and an average of 35 months.

**Table 9.** Image retractions by publisher.

| | Articles | ET (95% CI) | Median | Range | IQR |
|---|---|---|---|---|---|
| Elsevier | 281 | 50.40 (46.84–53.95) | 48 | 133 | 50 |
| Springer Nature | 132 | 35.20 (30.54–39.86) | 28 | 147 | 31 |
| Wolters Kluwer | 24 | 57.04 (40.12–73.97) | 63.5 | 127 | 79 |
| Wiley-Blackwell | 61 | 55.11 (46.39–63.84) | 51 | 127 | 60 |
| PLOS | 174 | 70.39 (66.40–74.37) | 73 | 128 | 33 |
| SAGE | 2 | 57.5 | | | |
| Taylor & Francis | 26 | 53.92 (40.74–67.11) | 43 | 126 | 46 |
| Hindawi | 38 | 50.76 (43.42–58.11) | 48.5 | 100 | 29 |
| Dove Medical Press | 44 | 19.36 (12.18–26.55) | 9 | 103 | 17 |
| E-Century Publishing | 2 | 22.5 | | | |
| Spandidos Publications | 21 | 22.57 (11.71–33.44) | 11 | 87 | 22 |

### 3.6. Plagiarism and Overlap

The total number of plagiarism and overlap cases for the top 11 publishers is 907 (893 unique articles from which 14 were retracted for both plagiarism and overlap): 509 for plagiarism and 397 for overlap. One of the publishers outperforms the others: with only 22 instances/21 articles representing less than 10% of their total retracted articles number, PLOS seems to have developed procedures that prevent the publication of articles that reuse text or plagiarize other scientific papers. However, post-publication average exposure time (ET) until retraction is the second largest of all publishers: 37.1 months.

Exposure time for plagiarism and overlap cases is presented in Table 10.

**Table 10.** Exposure time (ET) for plagiarism and overlap.

| | Plagiarim | Overlap | Cases | Articles | ET (95% CI) | Median | Range | IQR |
|---|---|---|---|---|---|---|---|---|
| Elsevier | 83 | 92 | 175 | 174 | 26.7 (23.43–29.97) | 20 | 114 | 26 |
| Springer Nature | 138 | 90 | 228 | 222 | 23.34 (20.92–25.75) | 19 | 100 | 24 |
| Wolters Kluwer | 86 | 54 | 140 | 139 | 24.08 (20.04–28.12) | 15 | 105 | 35 |
| Wiley-Blackwell | 30 | 47 | 77 | 77 | 25.17 (19.44–30.9) | 17 | 124 | 26 |
| PLOS | 11 | 11 | 22 | 21 | 37.1 (23.58–50.62) | 37 | 81 | 55 |
| SAGE | 23 | 20 | 43 | 41 | 22.83 (17.52–28.14) | 17 | 65 | 16 |
| Taylor & Francis | 29 | 38 | 67 | 66 | 23.14 (17.58–28.69) | 15.5 | 96 | 28 |
| Hindawi | 47 | 23 | 70 | 69 | 40.14 (33.02–47.27) | 42 | 151 | 50 |
| Dove Medical Press | 16 | 18 | 34 | 33 | 17.55 (10.7–24.4) | 10 | 84 | 23 |
| E-Century Publishing | 26 | 1 | 27 | 27 | 24.56 (17.04–32.08) | 22 | 63 | 34 |
| Spandidos Publications | 20 | 4 | 24 | 24 | 30.13 (20.89–39.36) | 33.5 | 82 | 40 |

The median values of ET are between 10 months (Dove Medical Press) and 42 months (Hindawi) with major publishers relatively well-positioned: 15 months for Wolters Kluwer, 15.5 months for Taylor & Francis, 17 months for Wiley-Blackwell and SAGE, 19 months for Springer Nature, and 20 months for Elsevier. Average values of ET spread between 17.5 months (Dove Medical Press) and 40.1 months (Hindawi).

Median values for the top three retraction reasons are represented in Figure 1.

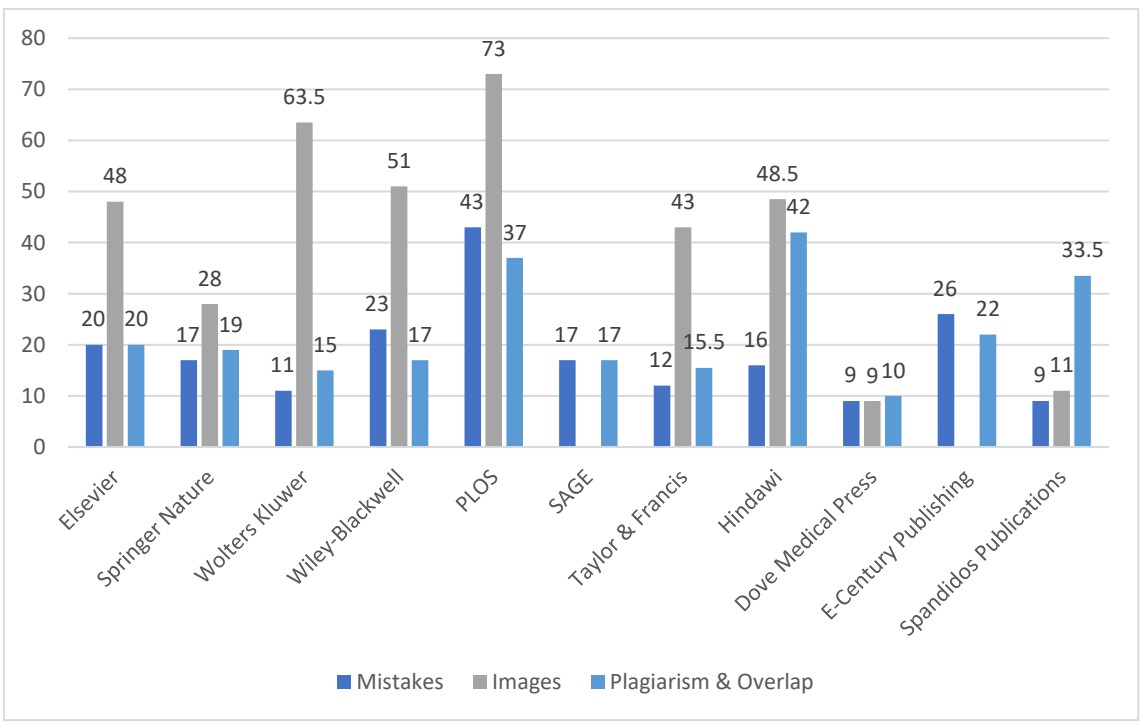

**Figure 1.** Median values (in months) of the top three retraction reasons for the top 11 publishers.

## 3.7. S(peed)D(etection)T(ransparency)P(recision) Score

To quantify the activity of the publishers, we calculated the SDTP score composed of six variables which, in our opinion, can provide an image of their involvement in ensuring the quality of the scientific literature. Baseline values of SDTP score components are shown in Table 11.

**Table 11.** Changes of main components of SDTP score for 2018–2020.

| | 2018 | 2019 | 2020 | |
|---|---|---|---|---|
| **S**peed (ET in months) | 24.65 | 26.74 | 28.89 | ▼ |
| **D**etection | 54.3% | 55.5% | 54.7% | ▲ |
| **T**ransparency—online article | 68.5% | 70.7% | 72.1% | ▲ |
| **T**ransparency—clear retraction note | 93.8% | 94.3% | 94.9% | ▲ |
| **P**recision—authors role | 12.8% | 11.9% | 10.5% | ▼ |
| **P**recision—no editorial errors | 95.7% | 95.8% | 96.3% | ▲ |
| ▼ = reduction ▲ = improvement | | | | |

The SDTP score of top 11 publishers and for the whole group of publishers below the 11th place was calculated for the intervals 2009–2020 (4844 articles, Table 12), 2009–2019 (3931 articles, Table 13), and 2009–2018 (3361 articles, Table 14),

**Table 12.** SDTP scores and rank for the period 2009–2020 (n = 4844). * Calculated for 4427 articles with more than one author and no editorial errors as retraction reasons.

| | | Score | Rank | Speed | Detection (REP) | Transparency | | Precision | |
|---|---|---|---|---|---|---|---|---|---|
| | | | | ET | | Online article | Clear retraction note | Authors role clear * | No Editor errors |
| **Baseline** | 4844 | | | 28.89 | 54.7% | 72.1% | 94.9% | 10.5% | 96.3% |
| **Elsevier** | 846 | | | 33.39 | 45.5% | 88.2% | 94.2% | 11% | 95.7% |
| | Points | −6.5 | 7 | −15.6 | −16.8 | 22.3 | −0.7 | 4.9 | −0.6 |
| **Springer** | 749 | | | 25.02 | 57.5% | 62.5% | 99.5% | 5.8% | 97.6% |
| | Points | −32.8 | 9 | 13.4 | 5.2 | −13.3 | 4.8 | −44.2 | 1.3 |
| **Wolters** | 346 | | | 23.4 | 66.2% | 70.8% | 93.4% | 6.2 | 93.1% |
| | Points | −7.4 | 8 | 19 | 21 | −1.8 | −1.6 | −40.7 | −3.3 |
| **Wiley** | 325 | | | 28.7 | 39.4% | 43.4% | 97.8% | 11.3% | 93.5% |
| | Points | −59.2 | 11 | 0.7 | −28 | −39.8 | 3.1 | 7.7 | −2.9 |
| **PLOS** | 275 | | | 57.76 | 80.4% | 100% | 100% | 5.9% | 99.3% |
| | Points | −49.8 | 10 | −99.9 | 46.9 | 38.7 | 5.4 | −44 | 3.1 |
| **SAGE** | 248 | | | 24.95 | 86.3% | 22.2% | 98.8% | 12.45 | 97.6% |
| | Points | 26 | 5 | 13.6 | 57.7 | −69.2 | 4.1 | 18.5 | 1.3 |
| **Taylor and Francis** | 223 | | | 23 | 65.9% | 39% | 98.2% | 19.8 | 93.3% |
| | Points | 82.9 | 1 | 20.4 | 20.5 | −46.9 | 3.5 | 88.5 | −3.1 |
| **Hindawi** | 140 | | | 37.18 | 72.9% | 99.3% | 99.3% | 10.4 | 100% |
| | Points | 50.2 | 2 | −28.7 | 33.2 | 37.7 | 4.6 | −0.5 | 3.9 |
| **Dove Medical Press** | 111 | | | 28.43 | 74.8% | 100% | 100% | 6.5% | 100% |
| | Points | 48.1 | 3 | 1.6 | 36.7 | 38.7 | 5.4 | −38.2 | 3.9 |
| **E−Century Publishing** | 71 | | | 18.18 | 78.9% | 100% | 100% | 0% | 74.6% |
| | Points | 2.9 | 6 | 37.1 | 44.2 | 38.7 | 5.4 | −100 | −22.5 |
| **Spandidos Publications** | 71 | | | 22.94 | 43.7% | 98.6% | 100% | 9.8% | 100% |
| | Points | 40.4 | 4 | 20.6 | −20.1 | 36.7 | 5.4 | −6.1 | 3.9 |
| **Rest of publishers** | 1439 | | | 25.74 | 48.6% | 75.4% | 89.2% | 13.5% | 97.2% |
| | | 27.9 | (5) | 10.9 | −11.2 | 4.6 | −6 | 28.7 | 0.9 |

**Table 13.** SDTP scores and rank for the period 2009–2019 (n = 3931). * Calculated for 3569 articles with more than one author and no editorial error as retraction reason.

| | | Score | Rank | Speed | Detection (REP) | Transparency | | Precision | |
|---|---|---|---|---|---|---|---|---|---|
| | | | | ET | | Online article | Clear retraction note | Authors role clear * | No Editor errors |
| **Baseline** | 3931 | | | 26.74 | 55.5% | 70.7% | 94.3% | 11.9% | 95.8% |
| **Elsevier** | 714 | | | 31.58 | 46.1% | 88.7% | 93.4% | 11.8% | 95.7% |
| | Points | −11.8 | 7 | −18.1 | −16.9 | 25.5 | −0.9 | −1.3 | −0.1 |
| **Springer** | 611 | | | 22.66 | 58.6% | 60.6% | 99.5% | 6.7% | 97.2% |
| | Points | −30.3 | 9 | 15.2 | 5.7 | −14.3 | 5.5 | −43.9 | 1.5 |
| **Wolters** | 300 | | | 21.54 | 67% | 70.3% | 92.7% | 7.1% | 93.7% |
| | Points | −4.5 | 6 | 19.4 | 20.8 | −0.6 | −1.7 | −40.2 | −2.2 |
| **Wiley** | 264 | | | 28.42 | 39% | 40.9% | 97.3 | 12.9% | 92% |
| | Points | −70.9 | 11 | −6.3 | −29.6 | −42.1 | 3.2 | 7.9 | −4 |
| **PLOS** | 199 | | | 50.02 | 74.4% | 100% | 100% | 8.2% | 99% |
| | Points | −34 | 10 | −87.1 | 34.1 | 41.4 | 6 | −31.7 | 3.3 |
| **SAGE** | 229 | | | 25.44 | 88.2% | 21.4% | 98.7% | 10.5% | 98.7% |
| | Points | −10.1 | 8 | 4.9 | 59.1 | −69.7 | 4.7 | −12.1 | 3 |
| **Taylor and Francis** | 159 | | | 21.67 | 62.3% | 27% | 97.5% | 28.1% | 91.2% |
| | Points | 103.7 | 1 | 19 | 12.3 | −61.8 | 3.4 | 135.6 | −4.8 |
| **Hindawi** | 109 | | | 28.82 | 68.8% | 99.1% | 100% | 12.5% | 100% |
| | Points | 71.5 | 3 | −7.8 | 24.1 | 40.2 | 6 | 4.6 | 4.4 |
| **Dove Medical Press** | 56 | | | 18.13 | 71.4% | 100% | 100% | 9.2% | 100% |
| | Points | 90.3 | 2 | 32.2 | 28.8 | 41.4 | 6 | −22.5 | 4.4 |
| **E−Century Publishing** | 54 | | | 13.67 | 85.2% | 100% | 100% | 0 | 66.7% |
| | Points | 19.5 | 5 | 48.9 | 53.6 | 41.4 | 6 | −100 | −30.4 |
| **Spandidos Publications** | 53 | | | 16.57 | 35.8% | 98.1% | 100% | 11.3% | 100% |
| | Points | 46.5 | 4 | 38 | −35.4 | 38.7 | 6 | −5.2 | 4.4 |
| **Rest of publishers** | 1183 | | | 25.17 | 47.3% | 75.7% | 88.3% | 15.3% | 96.7% |
| | | 20.9 | (5) | 5.9 | −14.6 | 7.1 | −6.4 | 28 | 0.9 |

The scores for 2018, 2019, and 2020 show signs of a consistent approach (such as Elsevier and Wiley-Blackwell, SAGE, Spandidos Publications), in which the increase in the number of retracted items is associated with an improvement in the overall score. There are also signs of a decrease in the quality of the retraction notes (such as PLOS, Wolters Kluwer) or the lack of noticeable changes (such as Springer Nature). Some publishers (Taylor & Francis, Hindawi, Dove Medical Press) seem to manage the quality control of their published articles more effectively, recording, however, a decrease of their overall scores between 2018 and 2020. The group represented by the rest of the publishers also marks an increase in the SDTP score. Individual results for the top 11 publishers and the "rest of the publishers" group are displayed in Tables 15–26 ( ● = performance degradation, ● = improvement of performance, ➡ = a difference of less than/equal to 0.1 points is considered stationary).

**Table 14.** SDTP scores and rank for the period 2009–2018 (n = 3361). * Calculated for 3037 articles with more than one author and no editorial errors as retraction reasons.

| | | Score | Rank | Speed | Detection (REP) | Transparency | | Precision | |
|---|---|---|---|---|---|---|---|---|---|
| | | | | ET | | Online article | Clear retraction note | Authors role clear * | No Editor errors |
| **Baseline** | 3361 | | | 24.65 | 54.3% | 68.5% | 93.8% | 12.8% | 95.7% |
| **Elsevier** | 602 | | | 29.99 | 45.5% | 88.5% | 92.5% | 12.6% | 95.3% |
| | Points | −8.3 | 8 | −21.7 | −16.2 | 29.2 | −1.4 | −2.2 | −0.4 |
| **Springer** | 538 | | | 21.88 | 59.1% | 58.2% | 99.4% | 7.2% | 97.8% |
| | Points | −30.7 | 10 | 11.2 | 8.9 | −15 | 6 | −44 | 2.2 |
| **Wolters** | 256 | | | 17.38 | 65.6% | 67.2% | 92.2% | 8.4% | 93.4% |
| | Points | 10.2 | 6 | 29.5 | 20.9 | −1.9 | −1.7 | −34.6 | −2 |
| **Wiley** | 246 | | | 26.87 | 39% | 39.4% | 97.2% | 12.9% | 92.3% |
| | Points | −78.1 | 11 | −9 | −28.1 | −42.5 | 3.6 | 0.5 | −3.6 |
| **PLOS** | 107 | | | 34.6 | 56.1% | 100% | 100% | 15.1% | 99.1% |
| | Points | 36.5 | 4 | −40.4 | 3.3 | 46 | 6.6 | 17.4 | 3.6 |
| **SAGE** | 218 | | | 25.89 | 89.4% | 21.6% | 98.6% | 10% | 99.5% |
| | Points | −21.9 | 9 | −5 | 64.7 | −68.5 | 5.1 | −22.2 | 4 |
| **Taylor and Francis** | 138 | | | 22.42 | 59.4% | 25.4% | 97.1% | 31.6% | 91.3% |
| | Points | 100.4 | 2 | 9 | 9.4 | −62.9 | 3.5 | 146 | −4.6 |
| **Hindawi** | 89 | | | 24.18 | 67.4% | 98.9% | 100% | 14.1% | 100% |
| | Points | 91.4 | 3 | 1.9 | 24.2 | 44.4 | 6.6 | 9.8 | 4.5 |
| **Dove Medical Press** | 42 | | | 18.86 | 71.4% | 100% | 100% | 12.5% | 100% |
| | Points | 109.4 | 1 | 23.5 | 31.5 | 46 | 6.6 | −2.7 | 4.5 |
| **E−Century Publishing** | 45 | | | 15.27 | 84.4% | 100% | 100% | 0% | 60% |
| | Points | 8.9 | 7 | 38.1 | 55.5 | 46 | 6.6 | −100 | −37.3 |
| **Spandidos Publications** | 43 | | | 14.26 | 34.9% | 97.7% | 100% | 9.3% | 100% |
| | Points | 31.8 | 5 | 42.1 | −35.8 | 42.6 | 6.6 | −27.6 | 4.5 |
| **Rest of publishers** | 1037 | | | 24.38 | 47.3% | 75.4% | 87.8% | 15.6% | 96.5% |
| | | 14.2 | (5) | 1.1 | −12.8 | 10.1 | −6.4 | 21.4 | 0.8 |

**Table 15.** Elsevier 2018–2020 change of SDTP score.

| **Elsevier** | **2018** | **2020** | |
|---|---|---|---|
| **Articles** | 602 | 846 | +40.5% |
| **S**peed | −21.7 | −15.6 | 🟢 |
| **D**etection rate | −16.2 | −16.9 | 🔴 |
| **T**ransparency—online article | 29.2 | 22.3 | 🔴 |
| **T**ransparency—clear retraction note | −1.4 | −0.7 | 🟢 |
| **P**recision—authors role | −2.2 | 4.9 | 🟢 |
| **P**recision—no editorial errors | −0.4 | −0.6 | 🔴 |
| **General** | −8.3 | −6.5 | 🟢 |
| **Rank** | 8 | 7 | |

**Table 16.** Springer Nature 2018–2020 change of SDTP score.

| Springer Nature | 2018 | 2020 | |
|---|---|---|---|
| Articles | 538 | 749 | +39.2% |
| **S**peed | 11.2 | 13.4 | 🟢 |
| **D**etection rate | 8.9 | 5.2 | 🔴 |
| **T**ransparency—online article | −15 | −13.3 | 🟢 |
| **T**ransparency—clear retraction note | 6 | 4.8 | 🔴 |
| **P**recision—authors role | −44 | −44.2 | 🔴 |
| **P**recision—no editorial errors | 2.2 | 1.3 | 🔴 |
| General | −30.7 | −32.8 | 🔴 |
| Rank | 10 | 9 | |

**Table 17.** Wolters Kluwer 2018–2020 change of SDTP score.

| Wolters Kluwer | 2018 | 2020 | |
|---|---|---|---|
| Articles | 256 | 346 | +35.1% |
| **S**peed | 29.5 | 19 | 🟢 |
| **D**etection rate | 20.9 | 21 | ⇨ |
| **T**ransparency—online article | −1.9 | −1.8 | ⇨ |
| **T**ransparency—clear retraction note | −1.7 | −1.6 | ⇨ |
| **P**recision—authors role | −34.6 | −40.7 | 🔴 |
| **P**recision—no editorial errors | −2 | −3.3 | 🔴 |
| General | 10.2 | −7.4 | 🔴 |
| Rank | 6 | 8 | |

**Table 18.** Wiley-Blackwell 2018–2020 change of SDTP score.

| Wiley-Blackwell | 2018 | 2020 | |
|---|---|---|---|
| Articles | 246 | 325 | +32.1% |
| **S**peed | –9 | 0.7 | 🟢 |
| **D**etection rate | −28.1 | –28 | ⇨ |
| **T**ransparency—online article | −42.5 | −39.8 | 🟢 |
| **T**ransparency—clear retraction note | 3.6 | 3.1 | 🔴 |
| **P**recision—authors role | 0.5 | 7.7 | 🟢 |
| **P**recision—no editorial errors | –3.6 | −2.9 | 🟢 |
| General | –78.1 | –59.2 | 🟢 |
| Rank | 11 | 11 | |

**Table 19.** PLOS 2018–2020 change of SDTP score.

| PLOS | 2018 | 2020 | |
|---|---|---|---|
| **Articles** | **107** | **275** | **+157%** |
| **S**peed | −40.4 | −99.9 | 🔴 |
| **D**etection rate | 3.3 | 46.9 | 🟢 |
| **T**ransparency—online article | 46 | 38.7 | 🔴 |
| **T**ransparency—clear retraction note | 6.6 | 5.4 | 🔴 |
| **P**recision—authors role | 17.4 | −44 | 🔴 |
| **P**recision—no editorial errors | 3.6 | 3.1 | 🔴 |
| **General** | **36.5** | **−49.8** | 🔴 |
| **Rank** | **4** | **10** | |

**Table 20.** SAGE 2018–2020 change of SDTP score.

| SAGE | 2018 | 2020 | |
|---|---|---|---|
| **Articles** | **218** | **248** | **+13.8%** |
| **S**peed | −5 | 13.6 | 🟢 |
| **D**etection rate | 64.7 | 57.7 | 🔴 |
| **T**ransparency—online article | −68.5 | −69.2 | 🔴 |
| **T**ransparency—clear retraction note | 5.1 | 4.1 | 🔴 |
| **P**recision—authors role | −23.2 | −25.4 | 🔴 |
| **P**recision—no editorial errors | −22.2 | 18.5 | 🟢 |
| **General** | **−21.9** | **26** | 🟢 |
| **Rank** | **9** | **5** | |

**Table 21.** Taylor & Francis 2018–2020 change of SDTP score.

| Taylor&Francis | 2018 | 2020 | |
|---|---|---|---|
| **Articles** | **138** | **223** | **+61.6%** |
| **S**peed | 9 | 20.4 | 🟢 |
| **D**etection rate | 9.4 | 20.5 | 🟢 |
| **T**ransparency—online article | −62.9 | −46.9 | 🟢 |
| **T**ransparency—clear retraction note | 3.5 | 3.5 | ⇒ |
| **P**recision—authors role | 146 | 88.5 | 🔴 |
| **P**recision—no editorial errors | −4.6 | −3.1 | 🟢 |
| **General** | **100.4** | **82.9** | 🔴 |
| **Rank** | **2** | **1** | |

**Table 22.** Hindawi 2018–2020 change of SDTP score.

| Hindawi | 2018 | 2020 | |
|---|---|---|---|
| **Articles** | **89** | **140** | **+57.3%** |
| **S**peed | 1.9 | −28.7 | 🔴 |
| **D**etection rate | 24.2 | 33.2 | 🟢 |
| **T**ransparency—online article | 44.4 | 37.7 | 🔴 |
| **T**ransparency—clear retraction note | 6.6 | 4.6 | 🔴 |
| **P**recision—authors role | 9.8 | −0.5 | 🔴 |
| **P**recision—no editorial errors | 4.5 | 3.9 | 🔴 |
| **General** | **91.4** | **50.2** | 🔴 |
| **Rank** | **3** | **2** | |

**Table 23.** Dove Medical Press 2018–2020 change of SDTP score.

| Dove Medical Press | 2018 | 2020 | |
|---|---|---|---|
| **Articles** | **42** | **111** | **+164.3%** |
| **S**peed | 23.5 | 1.6 | 🔴 |
| **D**etection rate | 31.5 | 36.7 | 🟢 |
| **T**ransparency—online article | 46 | 38.7 | 🔴 |
| **T**ransparency—clear retraction note | 6.6 | 5.4 | 🔴 |
| **P**recision—authors role | −2.7 | −38.2 | 🔴 |
| **P**recision—no editorial errors | 4.5 | 3.9 | 🔴 |
| **General** | **109.4** | **48.1** | 🔴 |
| **Rank** | **1** | **2** | |

**Table 24.** E-Century Publishing 2018–2020 change of SDTP score.

| E-Century Publishing | 2018 | 2020 | |
|---|---|---|---|
| **Articles** | **45** | **71** | **+57.8%** |
| **S**peed | 38.1 | 37.1 | 🔴 |
| **D**etection rate | 55.5 | 44.2 | 🔴 |
| **T**ransparency—online article | 46 | 38.7 | 🔴 |
| **T**ransparency—clear retraction note | 6.6 | 5.4 | 🔴 |
| **P**recision—authors role | −100 | −100 | ⇨ |
| **P**recision—no editorial errors | −37.3 | −22.5 | 🟢 |
| **General** | **8.9** | **2.9** | 🔴 |
| **Rank** | **7** | **6** | |

**Table 25.** Spandidos Publications 2018–2020 change of SDTP score.

| Spandidos Publications | 2018 | 2020 | |
|---|---|---|---|
| **Articles** | **43** | **71** | **+73.2%** |
| **S**peed | 42.1 | 20.6 | 🔴 |
| **D**etection rate | −35.8 | −20.1 | 🟢 |
| **T**ransparency—online article | 42.6 | 36.7 | 🔴 |
| **T**ransparency—clear retraction note | 6.6 | 5.4 | 🔴 |
| **P**recision—authors role | −27.6 | −6.1 | 🟢 |
| **P**recision—no editorial errors | 4.5 | 3.9 | 🔴 |
| **General** | **31.8** | **40.4** | 🟢 |
| **Rank** | **5** | **4** | |

**Table 26.** Group of publishers below 11th place 2018–2020 change of SDTP score.

| All Other Publishers | 2018 | 2020 | |
|---|---|---|---|
| **Articles** | **1037** | **1439** | **+38.7%** |
| **S**peed | 1.1 | 10.9 | 🟢 |
| **D**etection rate | −12.8 | −11.2 | 🟢 |
| **T**ransparency—online article | 10.1 | 4.6 | 🔴 |
| **T**ransparency—clear retraction note | −6.4 | −6 | 🟢 |
| **P**recision—authors role | 21.4 | 28.7 | 🟢 |
| **P**recision—no editorial errors | 0.8 | 0.9 | ➡️ |
| **General** | **14.2** | **27.9** | 🟢 |
| **Rank** | **5** | **5** | |

## 4. Discussion

There are many with the opinion that the actual number of articles that should be retracted is much higher than the current number [14,18,41–43].

Strengthening editorial procedures may decrease the number of articles retracted after publication. The quality and structure of the peer review process (author blinding, use of digital tools, mandatory interaction between reviewers and authors, community involvement in review, and registered reports) does have a positive role in preventing the publication of problematic articles [44]. However, at the moment, the process of correcting the scientific literature seems to be on an upward trend [11] which leads us to believe that there are still enough articles already published that require further analysis.

The post-publication analysis of scientific articles requires considerable effort from publishers/editors. Their performance when it comes to controlling the quality of the scientific product depends not only on internal (organizational) factors but also on external factors such as "post-publication peer review" or the intervention of authors/institutions [36,45] This dependence of quality control on external factors is also reflected in our data by the low involvement rate of the institutions to which the authors are affiliated: out of a total of 4844 retraction notes, only 465 (9.6%) mention the involvement of an institution. This number may be underestimated, but even so, given that editorial procedures sometimes include communication with authors' institutions, the lack of effective communication with

them makes the work of editors/publishers difficult when it comes to the quick retraction of an article or clarifying the retraction reasons.

Despite the impediments generated by the complexity of editorial procedures, if, as suggested [10], retracting an article may be regarded as a practical way to correct a human error, it would probably be helpful to measure publisher performance when it comes to quality control over scientific articles.

### 4.1. How Many Journals?

We note in our study a concentration of retracted articles in a relatively small number of publishers, the first 11 having 70.3% of the total retracted articles and 65.9% of the total number of scientific journals in which these are published. The top 45 publishers account for 88% of all articles and 79% of all journals. In our study, the total number of journals that have retracted at least one article is 1767, representing a small share of the 34,148 journals indexed in PubMed [46] (https://www.ncbi.nlm.nih.gov/nlmcatalog/?term=nlmcatalog+pubmed[subset], accessed on 9 January 2022) up to end of 2020.

Our study included only articles related to human health, excluding 775 articles that did not meet the inclusion criteria. Even if we added another 775 journals (assuming an article/journal), the share would remain extremely low, below 10%. On 9 January 2022, using the term "retracted publication (PT)" we get a total of 10,308 records starting with 1951. Using the same logic (one retracted article/journal) would result, in the most optimistic scenario, in another 5464 journals with retracted articles, which, added to the 1767 in our batch, would bring the total to 7231, just over 20% of the total number of journals registered in PubMed. However, we are helped here by a study published in 2021 [25], which analyzed 6936 PubMed-retracted articles (up to August 2019) and identified 2102 different journals, of which 54.4% had only one article retracted.

Our dataset contains, from September 2019–January 2021, 169 journals that retracted at least one article (within the included articles set) and 59 journals with at least one retracted study (within the excluded articles set). Taking into account these figures and including journal overlaps, we can say that the number of journals in PubMed that reported at least one retraction is at most 2330, less than 7% of the total number of active or inactive PubMed journals (the inactive status of a journal is not relevant for our estimation as the evaluation of the number of journal titles took into account the period 1951–2020, the time interval in which all journals had periods of activity).

Mergers, acquisitions, name changes, or discontinuations make it challenging to assess the number/percentage of journals affected by retractions from a publishers' portfolio. In the case of those with a smaller number of titles (PLOS, Spandidos, or Verducci), we see that post-publication quality control is implemented for almost all of their journals. When we talk about publishers with a medium number (>100) and a large number (>1000) of titles, the size of the quality problems of the published articles seems smaller. We are not sure if the data we found for an average publisher (out of 105 journals in the website portfolio [47] or 331 registered journals in PubMed or 94 active journal titles in Scopus, only 8 withdrew a total of 12 articles) or large (out of over 2000 of titles in PubMed only a little over 300 had articles retracted) reflect an effective quality control before publication or an insufficient quality control after publication.

Are over 90% of journals without a retracted article faultless? This is a question that is quite difficult to answer at this time, but we believe that the opinion that, in reality, there are many more articles that should be retracted [43] is justified.

### 4.2. Retraction Reasons

Of the 11 publishers analyzed, 9 recorded the highest number of retraction notes in 2020, which seems to reflect a growing interest in correcting the scientific literature.

### 4.3. Mistakes/Inconsistent Data

Detection of design and execution errors in research [48] may stop the publishing of the article and cause it to be rejected or corrected before publication. However, there are also situations in which the correction is necessary after publication, the errors not being detected in the peer review [49]. The leading retraction cause of scientific articles in our study is represented by mistakes/inconsistent data, with 1553 articles. The top 11 publishers have 1005 articles (64.7% of the total).

Data from the top 11 publishers show a large dispersion and an average ET duration of 15.4 months (Taylor & Francis), 17 months (Spandidos Publications), 19.7 months (Wolters Kluwer), and 41.6 months (PLOS). Median values, however, express better performances for most publishers (Table 5). At this time, we do not know if these values are due to delays in discovering errors, the length of the retraction procedure, or a systematic retroactive check implemented at the publisher or journal level.

The retraction notes mention 229 data fabrication cases, which justifies the need to develop and test the effectiveness of a set of statistical tools capable of detecting anomalies in published datasets (Hartgerink et al. 2019). In our opinion, the number of data fabrication cases may be underestimated: there are 293 cases in which researchers could not provide raw data, 180 cases of a lack of reproducibility, a lack of IRB approval in 134 cases, 47 cases of research misconduct, and 350 cases of fraudulent peer review. All can camouflage situations where data have been fabricated, even if this was not explicitly mentioned in the retraction. The first 11 publishers have 127/229 (55%) of the data fabrication cases.

### 4.4. Images

The images represent one of the retraction reasons, which has been growing in recent years, the increased interest in the subject highlighting its unexpected magnitude but also the development of tools that facilitate the detection of image manipulation in scientific articles [50–55].

The total number of image retractions is 1088. Of these, 805 (74%) belong to the top 11 publishers, and 587 of those (54%) belong to 3 publishers: Elsevier (281), PLOS (174), and Springer Nature (132)(see Table 9). Exposure time for articles retracted because of images is high for all publishers with two exceptions: Dove Medical Press (44 retractions, 19.3 months ET average, median 9 months) and Spandidos Publications (21 retractions, ET average 22.6 months, median 11 months). Springer Nature has a slightly better value (132 retractions, mean ET 35.2 months, median 28 months). The other publishers have values between 50 and 57 months, with medians between 43 and 63 months. PLOS has the highest ET: 70 months (median 73).

We reported in a previous study that 83% of image retractions were issued in the period 2016–2020 and the average value of ET is 49.2 months [11]. Therefore, the values mentioned above are not necessarily surprising, as they are the result of at least two factors: the relatively recent implementation at the editorial/publisher level of image analysis technologies (quite likely started between 2016–2018, see Tables 6–8) and the effort of publishers to analyze and retract from the literature articles published even 10–11 years ago. The good result of Dove Medical Press can be explained by the fact that most of the retracted articles are recently published and the retraction notes were given relatively quickly, in the period 2017–2020, being initiated by the publisher/editor in 33/44 cases. Fast turnaround times appear both at Spandidos Publications and at Springer Nature (Table 9). The percentages of initiation of the retraction by the publisher/editor are 50% and 56%, respectively. In these cases, it is possible that there is a workload that is easier to manage, shorter procedures/deadlines, or, simply, a better organization than competitors. The case of PLOS is a special one, the duration of 70 months of ET being 13 months longer than that of the penultimate place (Wolters Kluwer, 57 months); in 114/174 cases, the retraction was initiated at the editorial level. The profile of the published articles, the lack of involvement of the authors (only 25/174 cases involved the authors in one way or another), the slowness of the internal procedures, or the too-long time given to the authors to correct/provide

additional information and data could explain this value. The extended correspondence period with the institutions is not supported by our data (for the 30 cases in which the institution involvement was mentioned in the retraction note, the average value of the ET is 72 months).

The values recorded by the other publishers seem to reflect an effort to correct the literature but also difficulties in managing a rapid retraction process.

### 4.5. Plagiarism and Overlap (Text and Figures, No Images)

Detection of plagiarism/overlap can not only be an obligation of the authors/institutions to which they are affiliated [56] but should also be an essential component of scientific-product quality control at the publisher level. The identification of a plagiarized paper/an overlap case after publication represents, in our opinion, a modest editorial performance, especially in the context in which methods and applications (with all their shortcomings) are more and more widespread [57,58].

The total number of articles retracted for plagiarism/overlap is 1201. In 18 of them, both plagiarism and overlap are registered simultaneously. The top 11 publishers have 893 articles (74.3%). In terms of the number of articles, the first place is occupied by Springer Nature (222), followed by Elsevier (174) and Wolters Kluwer (139).

In our study, we identified only one publisher with a good performance in terms of quantity (PLOS—only 8% of all articles retracted were plagiarism/overlap). SAGE (17.4% of all retracted articles) and Wiley-Blackwell (23.7% of all retracted articles) also have reasonable levels.

Regarding the speed of retraction of plagiarism/overlap cases, the best performance is recorded by Dove Medical Press (average ET 17.5 months, median 10 months). The highest value of ET is recorded in Hindawi (40.1 months on average, median 42 months). Surprisingly, although it has a small number of articles (21), PLOS has the second-highest ET value (37.1 months average, 37 months median). The rest of the publishers have values around the average for the whole lot (average 24 months, median 17 months).

No publisher falls below an average ET of 22 months, with one exception. Their lower performance seems to be influenced, similar to mistakes or image retractions, by the late detection of a small number of articles (skewed distributions with a median lower than average). However, this modest performance shows severe problems at the editorial/publisher level. More than three years to retract a plagiarized/overlap article indicates significant gaps in publishers' detection and intervention capacities in this situation.

### 4.6. SDTP Score

The problem of correcting the scientific literature is one that, by the nature of the procedures to be followed, the resources to be allocated, and the complexity of the interactions needed to retract an article, sometimes exceeds the organizational capacity of a scientific journal [59–62]. We believe that publishers and editors' early and effective involvement in stopping, discouraging publication, and retracting QRP and QPP can help increase the quality of the scientific literature as a whole. The implementation of an independent evaluation system can help such an approach.

The parameters we use to evaluate the performance of publishers aim at the speed of the internal procedures of the journals in their portfolio (speed), the proactive behavior of the editorial staff (detection rate), and the transparency and precision of the retraction notes.

The interval 2018–2020 is characterized by an increase in the number of retracted articles by 44%, from 3361 to 4844: the increase varies between 164.3% (Dove Medical Press) and 13.8% (SAGE). At the entire dataset level, there is a decrease in performance in ET (an increase from 24.65 months to 28.89 months) and precision (the percentage of identification of responsible authors decreases from 12.8% in 2018 to 10.5% in 2020). The rest of the components are improving (Table 11).

For our dataset of PubMed retractions from 2009–2020, the first two places are held by two publishers of different sizes (Taylor & Francis and Hindawi), followed by another medium-sized publisher (Dove Medical Press). The same publishers appear, in changed order, when analyzing the periods 2009–2018 (Dove Medical Press, Taylor & Francis, Hindawi) and 2009–2019 (Taylor & Francis, Dove Medical Press, Hindawi).

We notice that an increase (between 2018 and 2020) in the volume of retracted articles has different effects at the publisher level: Dove Medical Press goes from first place to third place with a halving of the SDTP score (Table 23), PLOS (Table 19) goes from fourth place (2018) to tenth place (2020), and its score changes from a positive to a negative value. Taylor & Francis goes from second place in 2018 to first place in 2020 with a slight decrease in the score (Table 21), and Spandidos Publication improves its score (Table 25). Operating since the end of 2017 at Taylor & Francis (Taylor & Francis 2017), Dove Medical Press seems to have gained extra speed by almost doubling the number of retracted articles, perhaps because of access to more resources. PLOS's performance is affected by late retractions, the average ET increasing from 34.6 months in 2018 to 57.7 months in 2020.

The best and worst performances for the six parameters of the SDTP score are in Tables 27 and 28.

**Table 27.** Best SDTP score performance 2009–2020.

|  |  | Points |
|---|---|---|
| ET | E-Century Publishing | 37.1 |
| Detection | SAGE | 57.7 |
| Transparency (online article) | PLOS, Dove Medical Press, E-Century Publishing | 38.7 |
| Transparency (clear retraction notes) | PLOS, Dove Medical Press, E-Century Publishing and Spandidos Publications | 5.4 |
| Precision (authors role) | Taylor & Francis | 88.5 |
| Precision (no editorial errors) | Hindawi, Dove Medical Press, Spandidos Publications | 3.9 |

**Table 28.** Worst SDTP score performances 2009–2020.

|  |  | Points |
|---|---|---|
| ET | PLOS | −99.9 |
| Detection | Wiley-Blackwell | −28 |
| Transparency (online article) | SAGE | −69.2 |
| Transparency (clear retraction notes) | Spandidos Publications | −6 |
| Precision (authors role) | Springer Nature | −44.2 |
| Precision (no editorial errors) | Wolters Kluwer | −3.3 |

If we look at the individual performances of publishers between 2018 and 2020 (Tables 15–26), the picture presented is rather one of declining performance.

Compared to 2018, seven of the 11 publishers decreased their overall score in 2020. Only four publishers improve their performance between 2018 and 2020: Elsevier, Wiley-Blackwell, SAGE, and Spandidos Publications. The publishers below the 11th-place group also show an improvement in the score between 2018 and 2020.

Changes in score components between 2018 and 2020 show interesting developments:

- Only two publishers see an increase in four out of six indicators (Wiley-Blackwell and Taylor & Francis);
- The other publishers' group (below 11th place) also has an improvement in four indicators;

- Elsevier is the only publisher with three growing indicators;
- The other eight publishers register decreases to at least four indicators.

The changes go toward narrowing the gap between 1st place and 11th place: in 2018, the 11th place has $-78.1$ points, and 1st place has 109.4 points; in 2020, 11th place has $-59.2$ and 1st place 82.9 points.

The evolution of scores and indicators, often contrasting with the rank for 2020, associated with the narrowing of the gap between 11th and 1st place, leads us to anticipate a series of developments in the future:

- A possible improvement in the performance of the group of publishers below 11th place;
- A greater homogeneity of results for the first 11 publishers but also for the entire publishing environment;
- An improvement of the results for the big players;
- A continuation of the decrease of specific indicators (like ET) following the appearance of retractions for old articles in which the information necessary for a complete retraction note can no longer be obtained;
- Possible improvement for the involvement of publishers/editors or the editorial errors.

In this context, it is worth discussing whether the time required to retract an article reflects the publishers' performance or if there is a need for more complex measuring instruments that consider the multiple dimensions of publishing quality control.

### 4.7. Limitations

- A small number of errors (20) was discovered when analyzing PubMed records, concerning mainly the publication date or the retraction note date;
- The interpretation of retraction notes may generate classification errors, when several retraction reasons are mentioned;
- Modifications/completions made after the study by publishers or editors to the retraction notes on their sites may modify the figures obtained by us;
- The score is obtained by simple summation without taking into account the lower/higher weight that can be assigned to a specific component.

### 4.8. Recommendations for Future Work

The growing number of PubMed-retracted articles indicates the need for follow-up studies to see if new data confirm this trend.

There are variations between publishers of exposure time for articles retracted because of mistakes or data inconsistencies. A study on this subject may provide solutions for preventing the publication of such articles.

Image retractions are relatively new. For this reason, we consider it worthwhile to study in-depth the factors that determine the retraction of an article because of the images. Is it about technological progress? Have editorial policies already been changed, or are they currently being modified? Is there an editorial re-evaluation of the post-publication external peer review? Do the current retraction guidelines cope with image retraction difficulties?

The time required to remove plagiarized or overlap articles is long. Given the age of the anti-plagiarism technologies, it is not very clear what factors determine the further appearance of such articles in the scientific literature. A study that answers this question would probably allow for an improvement in editorial policies.

Studying changes to the SDTP score for publishers discussed in this article could help them implement consistent policies or correct issues that are more difficult to highlight at this time because of a lack of data.

## 5. Conclusions

*"Like a false news report, printed retractions do not automatically erase the error which often pops up in unexpected places in a disconcerting way. There is no instant "delete key" in science"* [63].

Retraction of problematic articles from the scientific literature is a natural process that should involve all stakeholders, including publishers.

Only a small number of journals indexed in PubMed are reporting retracted articles. We estimate that by January 2021, less than 7% of all journals in PubMed had retracted at least one article.

However, the correction efforts are obvious for all publishers, regardless of their size. Exposure time (ET), the involvement of publishers and publishers in initiating retractions, the online availability of retracted articles, and specifying the responsibility of authors are aspects that can be improved for all publishers reviewed in this paper.

The clarity of the retraction notes and the editorial errors are two indicators for which the potential for progress is limited only to specific publishers. The COPE guidelines must not only be accepted but must also be implemented. In this context, we believe that introducing a reporting standard for retraction notes will allow, along with the introduction of new technologies and the exchange of information between publishers, and better quality control of the scientific literature, one that can be easily measured, reproduced, and compared. The SDTP score proposed by us is only a small step in this direction.

**Author Contributions:** Conceptualization, C.T., L.P. and B.T.; Data curation, C.T. and B.T.; Methodology, C.T.; Project administration, C.T.; Software, C.T.; Validation, L.P.; Writing—original draft, C.T.; Writing—review & editing, L.P., B.T. All authors have read and agreed to the published version of the manuscript.

**Funding:** This research received no external funding.

**Data Availability Statement:** The data that support the findings of this study are available on request from the corresponding author.

**Conflicts of Interest:** The authors declare no conflict of interest.

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
