# Peer review of "Correction of the Scientific Production: Publisher Performance Evaluation Using a Dataset of 4844 PubMed Retractions"

_publications, doi:10.3390/publications10020018_

Round 1
Reviewer 1 Report
The manuscript needs revision before acceptance.
- Table 1 is not complete.
- Clear objectives must be provided.
- What are the recommendations for future work?
- Authors stated that "The methodology used to collect the data is presented in detail in another article (Toma and Padureanu
2021)". At least a brief description is needed. - Change the presentation style as per the journals' guidelines.
Reviewer 2 Report
This study investigates retraction notes and reasons for retraction in different publishers. Authors use the same data set that was used in another recently published manuscript in the open journal of social sciences.
General comment:
- Use terminology consistently: either use withdrawn articles or retracted articles.
- Citations: please avoid citing several resources after a general statement, e.g., you cite 14 references in the introduction the general statement "numerous articles making a valuable contribution in this field". You have to indicate the contribution of each, otherwise this is a meaningless citation and an instance of citation padding which is unethical.
- Structure: please improve the structure according to journal guidelines. e.g., limitations do not belong to the introduction section, results do not belong to the discussion section.
In what follows, I provide feedback per section:
Introduction:
- I advise authors to provide a more meaningful introduction and present the rationale for their study in a more concrete manner. The intro starts with a long quotation followed by how much money publishers make. Then authors claim that because publishers make a lot of money and have a monopoly, they should prioritise improving the quality of publications. This is a flimsy and inaccurate rationale. So many businesses across all sectors make money but they are under no obligation to improve quality. The fact that so many publishers are involved in this industry also implies that there is no monopoly; authors can always publish with other non-profit journals too.
- Authors note "there is little information on the article retractions at a publisher level", this claim is neither (because a simple Google search shows some studies that did this) accurate nor properly substantiated.
- Authors suggest a score but it is not clear what has informed the development of this score. It is true that authors cite COPE but why should these indicators be used in a score? Why do we need a score at all? Later they call this score a tool, which is confusing.
Metodology
- Each article should clearly explain its methods. So please add a summary of used methods here.
- The current use of the term detection in the STDP score is a bit strange because we can only know the detection rate once we have information about the total number of possible fraudulent papers. Given that a lot of fraudulent papers are never retracted or discovered, this terms seems to be a misfit.
Results
I have no statistical knowledge so I can't provide feedback on used tests etc. My comments are more general here:
- Authors note "These data show an increased efficiency in dealing with image issues" but this is largely hypothetical. We do not know how many papers have image issues, so neither an increase nor a decrease in the total number of retracted articles could inform us about efficiency.
- It is not clear why table. 11 only presents the data from 2018 to 2020 and not for other years.
- In tables 12, 13 and 14 authors apply their suggested SDTP score to three different intervals but it is not clear why they do this.
- The caption of tables 12, 13 and 14 notes "articles with more than one author and no editorial error as retraction reason", but the definition of detection in table 1. reads: "Percentage of total withdrawn articles for which the withdrawal was initiated/involved the editor, the editorial board, the publisher or institutions, without authors". This is really confusing.
- Table 15-26, only presents the data from 2018 to 2020 and not for other years. Comparing two data points from 2018 and 2020 does not provide enough information so that we could call it evolution..
Discussion
This is still a weak section. I suggest authors to remain focused on publishers and e.g., what they can do to foster and incentivise post-publication reviews?
- Most of what is presented in the subsection entitled "How many journals" belong to the results section.
- Authors note "in the case of those with smaller number of titles (PLOS ...) it is easy to realize that quality problems affect all ... ", which is vague. What is exactly easy here? Also, PLOS is a mega journal with an enormous number of published articles. Perhaps using the number of titles as a unit of analysis is not really useful here. Further in the same paragraph, authors make more vague statements about quality control. I am not sure what authors mean here.
- Authors cite the view that "there are many more articles that should be retracted... covered by actual figures", but they don't provide the justification: what do you mean with actual figures? Whose figures? Where are these figures?
Reviewer 3 Report
In the abstract and possibly later in the study, it is worth correcting the term withdrawal to retraction in all cases. Withdrawal is typically a request from the author that occurs before the stage of publication. The authors in this study, on the other hand, actually deal with retraction, which can be initiated by both authors and editors (or by the publisher) and takes place after publication.
The authors have observed a huge amount of data and made a very important analysis that shows what publishing trends can be detected nowadays as a result of the pressure on researchers. These are important findings as publishers can prepare for it and so the purity of science can be maintained. The authors also highlight an indicator that shows how many articles each publisher had retracted. I do not recommend publishing this in its form, because there is a huge difference in size among the publishers (how many journals are managed by each of them), so this indicator is not suitable for any conclusions. Table 2 would also be really correct if it included the number of articles published by each publisher during the period considered. If we compare the number of retracted articles to this, we can deduce to the quality of the review processes.
Table 3 requires a more detailed explanation even if the authors have no information as to why publishers do not use a high proportion of retraction notes. After that, the study contains a lot of valuable tables, the explanation of which, however, is superficial or missing. This is partly solved from page 20, but needs to be expanded.
I don’t know why the authors didn’t use the journal’s template, but it’s definitely worth submitting the corrected study using that.
Otherwise it is an excellent work, and may be suitable for acceptance after minor changes.
Round 2
Reviewer 2 Report
N/A